# Epidemiology of Traditional Cigarette and E-Cigarette Use Among Adolescents in Poland: Analysis of Sociodemographic Risk Factors

**DOI:** 10.3390/ijerph21111493

**Published:** 2024-11-10

**Authors:** Paulina Kurdyś-Bykowska, Leon Kośmider, Wojciech Bykowski, Dawid Konwant, Krystyna Stencel-Gabriel

**Affiliations:** 1Department and Clinical Department of Pediatrics in Bytom, Medical University of Silesia, 40-055 Katowice, Poland; dawidkonwant@interia.pl (D.K.); kgabriel@sum.edu.pl (K.S.-G.); 2Faculty of Science and Technology, Jan Długosz University in Częstochowa, Armii Krajowej 13/15, 42-200 Czestochowa, Poland; l.kosmider@ujd.edu.pl; 3MTZ Clinical Research Powered by Pratia, ul. Gładka 22, 02-172 Warszawa, Poland; wojciech.bykowski@pratia.com

**Keywords:** e-cigarettes, electronic cigarettes, smoking, tobacco, students, adolescents

## Abstract

Background: E-cigarettes are electronic nicotine-dispensing systems in the form of an aerosol. Their popularity among adolescents is growing at an exceedingly fast pace. Methods: This study aimed to determine the prevalence of the use of traditional cigarettes and e-cigarettes and identify demographic risk factors for the use of these products by adolescents in large and small cities and rural areas in Poland. This cross-sectional study conducted in 2021 aimed to assess the prevalence of traditional cigarette and e-cigarette use among adolescents aged 12–18 in Poland, while identifying demographic risk factors associated with their usage. A total of 10,388 adolescents participated, predominantly from rural areas. Results: Findings revealed that 12.3% were traditional cigarette smokers, with 90% smoking in the past month, while 14.9% were e-cigarette users, with 84.7% using them in the past month. Dual users accounted for 6.4% of respondents. Non-smokers were younger, and e-cigarette users were more likely to be boys from larger cities. Moreover, mothers of non-smokers tended to have higher education levels than those of traditional cigarette smokers. Conclusions: This study provides important new insights into demographic predictors associated with the use of specific devices, which can help inform targeted interventions to reduce e-cigarette use.

## 1. Introduction

Electronic cigarettes (e-cigarettes) are battery-powered electronic nicotine delivery systems that generate an aerosol containing nicotine [1]. Scientific research on the use of e-cigarettes is carried out in many parts of the world. Initially, most researchers assessed the use of e-cigarettes in a group of adults, focusing primarily on assessing the effectiveness of e-cigarettes as a smoking cessation tool. Through analysis of the experience and outcomes of those who used the devices to reduce traditional smoking, researchers sought to understand how e-cigarettes impact the tobacco cessation process [2,3]. They consider the influence of simultaneous use of electronic nicotine delivery systems (ENDSs) and smoked tobacco products, the impact of the liquids used on reducing exposure to toxic substances, and the effect on the users’ health [4,5]. Studies have also covered the analysis of the effect of e-cigarettes, in particular the liquid components, on the respiratory system, heart and other body systems [6,7]. Conducting similar analyses in adolescents is more complicated; hence, the authors focus on slightly different research aspects compared to the adult population. First, the authors assess the reasons for the growing popularity of e-cigarettes among children and adolescents and conduct analyses of trends related to their use. In addition, researchers carefully analyzed the increase in the use of e-cigarettes in various age groups in children and adolescents, simultaneously assessing factors that may influence the process of making decisions regarding the use of these products [8,9]. Research conducted on the adolescent population also focuses on a phenomenon known as the “gateway effect”, which refers to a situation where teenagers or adolescents who start using e-cigarettes later switch to traditional tobacco cigarettes. This happens through exposure to the nicotine contained in e-cigarettes, which are often advertised as less harmful and, therefore, may create the illusion that using them is not as risky or harmful as smoking traditional cigarettes. The gateway effect also involves becoming used to smoking rituals. E-cigarettes often come in a variety of flavours, which appeal to adolescents. After becoming familiar with inhaling active substances using e-cigarettes, some users may easily switch to traditional cigarettes [10,11,12]. Another aspect is marketing and advertising. Researchers analyze the marketing strategies of companies producing e-cigarettes, with particular emphasis on their impact on adolescents. Doubts about whether e-cigarette advertising is directed towards children and adolescents is another important element of the conducted analyses [13,14]. Researchers also investigate the impact of using e-cigarettes on children and adolescents, focusing heavily on the consequences for the respiratory system. Additionally, researchers examine the chemical ingredients used in e-cigarettes in an attempt to identify potential health risks resulting from their use [15,16].

The most important, harmful-to-health and highly addictive ingredient of e-cigarette liquid is nicotine. This substance is a stimulant affecting the central nervous system, which binds to nicotinic acetylcholine receptors in the brain. The binding to these receptors causes dopamine release, which is involved in producing pleasant sensations. Nicotine is highly addictive, and stopping its use may lead to withdrawal symptoms. People who start using nicotine at a young age face a greater risk of developing health problems in adulthood. These problems include heart disease, lung disease, cancer, and increased susceptibility to various infections [17,18]. Exposure to nicotine during childhood can also affect the brain, causing critical changes in processes related to learning, memory, and mood [19]. This is why this topic is crucial for adolescents’ health and development.

The GYTS (Global Youth Tobacco Survey) report indicates that adolescents in Poland occupy a leading place in terms of e-cigarette use (a result of 23.4% in 2016), followed, in order, by students from Ukraine (18.4% in 2017) and Latvia (18.0% in 2019) [20]. In the context of the widespread popularity of e-cigarettes and the associated health and social impacts, it is crucial to identify the group of adolescents at risk in order to effectively prevent them from starting to use or experiment with these products.

This study aimed to determine the prevalence of the use of traditional cigarettes and e-cigarettes and identify demographic risk factors for the use of these products by adolescents in large and small cities and rural areas in Poland.

## 2. Materials and Methods

This study presents an analysis of data on tobacco use among adolescents obtained from a survey conducted in primary and secondary schools in all voivodeships in Poland.

After analyzing the PATH (Population Assessment of Tobacco and Health) study and its findings, we independently developed a self-designed questionnaire that was used in this study [21]. The survey was planned for the 1st half of 2021, i.e., during the SARS-CoV-2 pandemic, when classes in schools across Poland were held exclusively remotely. The survey questionnaire was made available to adolescents on the SurveyMonkey platform and was sent to school Head Teachers to familiarize educators and teachers with its content. Subsequently, school Head Teachers across Poland were asked to make the questionnaires available to adolescents during school hours for completion. Since the questionnaire was anonymous, it did not require the collection of additional consent from the survey participants, and participation in the study was voluntary. In order to ensure the complete anonymity of the students surveyed, no data about the school from which the responses came were collected.

The questionnaire consisted of 82 questions divided into 5 parts. The sections were devoted to demographic data, smoking, use of electronic cigarettes, prevalence of respiratory symptoms and data on the COVID-19 pandemic. This publication involved the analysis of responses to questions from the first three sections of the survey. Demographic data included gender, age, the type of school the students attended, the province and the size of the locality they lived in. The survey also included questions about adolescents’ physical activity, such as whether they attended PE lessons, whether they exercised regularly during PE lessons, and whether they attended training sessions at sports clubs. The adolescents were also asked about their parent’s educational background and their smoking and electronic cigarette use habits. The next section included questions about the initiation of the use of different forms of nicotine by the adolescents, the first nicotine-containing product, and the circumstances surrounding it. Then, the adolescents answered questions about their habits related to traditional and electronic cigarettes and the health symptoms they experienced. The survey was approved by the Bioethics Committee of the Medical University of Silesia in Katowice PCN/0022/KB1/63/21.

The survey was initially provided to a selection of schools for the 24 h pilot, during which we collected 700 responses. This effort allowed the authors to move into the validation stage, following a carefully structured methodology to ensure the survey instrument’s reliability and validity. Our first step was to confirm face validity, ensuring the survey items were relevant and appropriately aligned with the constructs we intended to measure. Conducting a pilot test allowed us to gauge the clarity of the questions, identify any areas of confusion, and address ambiguities within the response options. Once we collected the pilot test results, we cleaned the data to remove incomplete or inconsistent responses, which provided a solid foundation for further analysis. We then ran statistical tests, including Principal Component Analysis (PCA), to identify any underlying dimensions and assess the suitability of each item in capturing our intended constructs. To further evaluate the internal consistency of the survey, we calculated Cronbach’s alpha (α > 0.7). This metric gave us insights into how well the questions within each section worked together to measure a cohesive concept. Based on the findings from PCA and Cronbach’s alpha, we revised certain survey items to strengthen their reliability and validity. By following this thorough validation process, we aimed to enhance the quality of the data collected, increase our confidence in interpreting the findings, and extend them to a broader student population. This systematic approach not only strengthens the credibility of the research but also lays a solid groundwork for the subsequent analyses and conclusions drawn from the survey.

The survey was conducted among students in grades 6–8 of primary school and grades 1–4 of secondary schools, aged 12–18. A total of 13,474 responses were obtained, of which 10,388 were correctly completed. The overall response rate was 77%.

A total of 10,388 adolescents participated in the survey, 55.6% (n = 5778) of whom were girls and 44.4% (n = 4610) of whom were boys. The median age was 16.16 years (SD = 1.70). Over ½ (n = 5492, 52.9%) of the adolescents lived in rural areas, almost ¼ (n = 2573, 24.8%) of them lived in small towns, 1182 lived in big cities (11.4%), and 1141 (11%) lived in medium-sized towns. The educational level of the surveyed adolescents’ parents was analyzed. The question was discretionary since the adolescents may not know what education their parents have.

Other demographic data (parents’ education, type of school and details about adolescents’ physical activity) are presented in Table 1.

## 3. Statistical Analyses

Statistical analyses were performed using the IBM SPSS Statistics software, version 25. Descriptive statistics were computed to characterize central tendencies, variability, and distributional characteristics of the dataset. The Kolmogorov–Smirnov test was used to assess the normality of data distributions. For inferential analysis, the Mann–Whitney U test was applied to evaluate differences between two independent groups, while the Kruskal–Wallis test was used to compare differences across multiple groups. Spearman’s rank correlation coefficient (ρ) was calculated to assess the magnitude and direction of associations between ordinal variables or continuous variables that did not meet normality assumptions. Furthermore, chi-squared (χ^2^) tests, with Bonferroni adjustment for multiple comparisons, were utilized to analyze associations between categorical variables. A significance level of α = 0.05 was established as the criterion for statistical significance throughout the analyses.

## 4. Results

### 4.1. Tobacco Use Initiation

The initial part of the survey included questions designed to characterize adolescents’ first contact with tobacco products. The status of ever using tobacco products was given to a adolescent who answered “yes” to the question “Have you ever used such products as traditional cigarettes, e-cigarettes, heated tobacco products (e.g., IQOS), or marijuana?” Such responses were given by 4367 adolescents (42.04%). Of the total sample of 10,388 surveyed, 3347 adolescents (32.2%) declared that they had smoked cigarettes, 3924 (37.8%) had used e-cigarettes, 1078 (10.4%) had tried a heated tobacco product (HTP) and 1309 (12.6%) had tried marijuana. In total, 6021 (57.96%) of the respondents declared that they had never smoked any of these products. Next, the analysis covered the age and tobacco product used for the first time, assessed through the questions “Which of the above did you use first?” and “How old were you then?” In total, 21.4% of the adolescents declared that the first product they had used was traditional cigarettes; 19.4% tried e-cigarettes first, 1% tried marijuana and 0.3% tried an HTP. The median age of reaching for the first tobacco product was 14.33 years (SD = 2.07). The reason for trying the product and the place where the first attempt had taken place were determined through the questions “Why did you start smoking?” and “In what situation did you smoke cigarettes or other products mentioned earlier for the first time?” The majority of the adolescents who declared that they had tried any of these products at least once claimed the most common moment to try tobacco products for the first time was after school with their friends (n = 2170, 49.7%). At the same time, the main motive was curiosity (n = 2850, 65.3%). The surveyors also wanted to determine the locations of the adolescents’ habitual use of a particular product. They could choose multiple locations from a list containing places at school and at home. Most adolescents use these types of products at parties (n = 3361, 77%). A large percentage of respondents use these products at home (n = 1590, 36.4%) with a relatively large share of them smoking near school (n = 1766, 40.4%) or at school (n = 914, 20.9%). Nearly ten percent of all smoking adolescents smoke tobacco products in front of their parents (n = 434, 9.9%).

### 4.2. Use in the Past 30 Days

In a further analysis, the authors decided to consider those who had used either traditional cigarettes or e-cigarettes in the past 30 days. The adolescents who gave an affirmative response to the question “Have you used traditional cigarettes in the past 30 days?” were included in the current tobacco smoker group (n = 1275). At the same time, the adolescents who answered “yes” to the question “Have you used e-cigarettes in the past 30 days?” (n = 1546) were categorized as current e-cigarette users. In most works, researchers compare e-cigarette users with adolescents using traditional cigarettes [22,23]. However, numerous tobacco users tend to use more than one tobacco product at the same time [24,25,26]. Therefore, the initial analysis covered the first two groups, and later in the study, users of both products were distinguished as a separate group, named “dual users”.

Table 2 presents data categorizing adolescents into two groups: those who smoke traditional cigarettes and those who use e-cigarettes.

In the case of traditional cigarettes, 1275 adolescents, representing 90% of all traditional cigarette smokers (12.3% of the total sample), claimed to have smoked in the past 30 days. At the same time, in the case of e-cigarettes, 84.7% of the users had used this type of device in the past 30 days (n = 1546, 14.9% of the total sample). Moreover, adolescents using both traditional and electronic cigarettes, so-called “dual users”, accounted for 6.4% of all the respondents.

The median age of adolescents smoking cigarettes was M = 16.95 years (SD = 1.20). Most respondents lived in rural areas (n = 651, 51.1%). As for the respondents’ fathers, the largest percentage had vocational education.

As for adolescents using e-cigarettes, the median age in this group was M = 16.98 years (SD = 1.23), and most of them lived in rural areas (n = 713, 46.1%).

### 4.3. Methods of Purchase

Adolescents were asked how they obtained their e-cigarettes. The most common way of obtaining traditional cigarettes was to buy them themselves, which is the answer given by 846 respondents (66.4%). Similarly, in the group using e-cigarettes, respondents bought e-cigarettes themselves from a store (n = 986, 63.8%). Given that some respondents were of legal age at the time of completing the survey, a comparison was made between how traditional cigarettes and e-cigarettes were purchased among those under and over 18. Adult users were more likely to buy their own cigarettes and e-cigarettes from a store and were less likely to use other techniques (Table 3 and Table 4). The adolescents were also asked whether they concealed that they smoked cigarettes or used e-cigarettes from their parents. In total, 728 (57.1%) of the adolescents currently smoking traditional cigarettes indicated that they concealed cigarette smoking from their parents, with 104 (8.2%) indicating that they hid that fact only from their father and 65 (5.1%) only from their mother. A total of 826 (53.4%) e-cigarette users concealed the fact that they use these devices from their parents; 628 respondents (40.6%) hid the fact from both parents, 54 adolescents (3.5%) only from their mother, and 141 (9.1%) only from their father.

### 4.4. E-Cigarette Type

E-cigarettes are a convenient option due to their small size, ease of use, and availability in a variety of flavours [27,28]. In the survey, the adolescents answered the question “What type of e-cigarette do you use?” They could choose by clicking on the photo of the e-cigarette that was most similar in appearance to the one they use themselves. Since adolescents may own more than one device, it was a multiple-choice question. The most commonly chosen type of e-cigarette was the mod box, indicated by 66.8% of the respondents; 34.2% of those surveyed used pod e-cigarettes. E-cigarettes can be classified into two main types: open and closed systems. The first are e-cigarettes that allow the user to refill the e-liquid tank or cartridge with the e-liquid of their choice. They provide greater flexibility and a wider range of flavour options, and are also known as refillable e-cigarettes. Closed systems are e-cigarettes that use pre-filled cartridges or are equipped with capsules that cannot be refilled. They are also known as disposable e-cigarettes or pod systems. E-cigarettes with closed systems have limited flavour options and nicotine levels as they are already pre-filled. In total, 1455 (94.1%) surveyed adolescents used an open system. The remaining 91 (5.9%) claimed they used closed-system e-cigarettes; 44.4% of current e-cigarette users use ready-made liquids and 43.5% choose premixes, i.e., liquids without nicotine, to which they add the liquid with nicotine themselves, while 12.5% of the users prepare their liquid themselves from available components.

### 4.5. Nicotine

Nicotine levels in e-cigarettes can vary, depending on the product and user preference. Nicotine levels are usually measured in milligrams per milliliter (mg/mL) or as a percentage (%). A typical nicotine level in e-liquids ranges from 0 mg/mL to 36 mg/mL or from 0% to 2%; 0 mg/mL or 0% is considered a nicotine-free level. A range of 3–6 mg/mL or 0.3–0.6% is generally considered low. A level of 9–12 mg/mL or 0.9–1.2% is considered medium, while 16–20 mg/mL or 1.6–2.0% is considered high. The current e-cigarette users were asked what level of nicotine they used in their devices. They most often used low nicotine levels (3–6 mg). This answer was given by 588 adolescents (38%); 392 adolescents (25.4%) claimed they used e-cigarettes with medium nicotine levels (9–12 mg), while 352 of them (22.8%) declared they used e-cigarettes with high nicotine levels (22.8%). The remaining 214 adolescents used e-cigarettes without nicotine (13.8%).

### 4.6. Flavours

Flavoured cigarettes were banned in Poland on 20 May 2020. The ban applies to the sale, distribution and promotion of cigarettes with any type of flavour, such as fruit, candy, mint and others. Accordingly, questions about flavours referred only to e-cigarette users, as flavoured liquids are still available in Poland. Adolescents are most likely to use fruit flavours (n = 902, 58.3%) and menthol flavours (n = 202, 13.1%).

### 4.7. Demographic Variables Versus Tobacco Use

In a further analysis, an additional group was distinguished among the adolescents declaring the use of both traditional cigarettes and e-cigarettes. The group was composed of the adolescents using both forms of nicotine supply simultaneously (n = 667), and was called “dual users”.

The comparison concerned the adolescents who were non-smokers (n = 8234), traditional cigarette smokers (n = 608), e-cigarette-only users (n = 879) and dual users (n = 667), presented in Table 5. The non-smoking adolescents were younger than those in the other three groups (*p* < 0.001). The adolescents in the dual-user group were younger than those using only cigarettes and those using only electronic cigarettes. The proportion of boys was higher in the group of e-cigarette users compared to those who had not used tobacco products in the past 30 days (*p* < 0.001). Other differences were not statistically significant. The adolescents using electronic cigarettes lived in larger cities compared to the those in other groups ((1) *p* < 0.001, (2) *p* = 0.001 and (4) *p* = 0.047). Considering parents’ education, mothers of the non-smoking adolescents had a higher level of education compared to mothers of the adolescents smoking traditional cigarettes (*p* < 0.001) and to mothers of those using both forms of nicotine delivery (*p* = 0.009). At the same time, mothers of the adolescents using only e-cigarettes had a higher level of education than mothers of the smoking adolescents *(p* = 0.008). Fathers of the smoking adolescents had a lower level of education compared to fathers of the adolescents in the other groups ((1) *p* < 0.001, (3) *p* = 0.001 (4) *p* = 0.011). The proportion of smoking parents was lower in the group of non-smoking adolescents compared to the other groups ((2) *p* < 0.001, (3) *p* < 0.001 and (4) *p* < 0.001). The proportion of smoking parents was also lower in the group of the adolescents using electronic cigarettes compared to the group of the smoking adolescents (*p* = 0.004). The researchers also wanted to know whether there was a relationship between parents smoking traditional cigarettes and adolescents smoking. The proportion of parents smoking traditional cigarettes was lower in the group of the non-smoking adolescents compared to the other groups ((2) *p* < 0.001, (3) *p* < 0.001 and (4) *p* < 0.001). Parents of the adolescents using only e-cigarettes smoke traditional cigarettes less frequently than parents of the smoking adolescents (*p* < 0.001). The analysis also covered the fact of using e-cigarettes by the respondents’ parents. Parents of the non-smoking adolescents used these devices less frequently compared to parents of the adolescents who smoked cigarettes (*p* < 0.001) and those who used e-cigarettes (*p* < 0.001). In contrast, parents of the adolescents who used both products used electronic cigarettes more often than parents of the adolescents who smoked only cigarettes and those who used only e-cigarettes.

The proportion of primary school students was significantly higher in the group of non-tobacco users compared to the other three groups, with the largest number of students using e-cigarettes in secondary technical schools. The sports profile of the school did not correlate with the fact that different tobacco products are used. The non-smoking adolescents were more likely to exercise during PE lessons compared to their peers in the other groups ((2) *p* < 0.001, (3) *p* = 0.001 and (4) *p* < 0.001). At the same time, the adolescents who smoked only traditional cigarettes were more likely to participate in these lessons compared to the dual users. The non-smoking students were more likely to exercise regularly outside of school, compared to the smokers (*p* < 0.001) and dual users (*p* < 0.001). In contrast, the adolescents who use e-cigarettes exercised more regularly than peers who smoked traditional cigarettes (*p* = 0.003) and dual users (*p* = 0.035). More advanced forms of sports participation, such as regular training at sports clubs, did not associate with cigarette smoking or the use of electronic cigarettes.

## 5. Discussion

The study analyzed survey data from a large sample of 10,388 primary and secondary school students from all over Poland. It focused on the prevalence and habits of using tobacco products, and e-cigarettes in particular, among adolescents in Poland.

In Poland, the percentage of children and adolescents currently using e-cigarettes has been steadily increasing over recent years. In 2010–2011, it was 2%, rising to 8% in 2013–2014 and to 11% in 2015–2016. In 2016, 28% of boys and 18.6% of girls aged 13–15 claimed that they were users of e-cigarettes [29]. Similarly, the frequency of e-cigarette and traditional cigarette use increased simultaneously from 4% in 2010–2011 to 24% in 2015–2016 [30]. In our study, the frequency of use of e-cigarettes in the last 30 days among adolescents is slightly higher at 14.9%, while dual users accounted for slightly less—6.4% of the study population. In the 2022 GYTS Poland report, 22.3% of surveyed children reported current e-cigarette use [20]. However, the observed differences in the results may be attributed to the age groups covered by the studies; while the GYTS survey included children aged 13–15, our study applied broader age criteria, encompassing children aged 12–18. This reveals that the characteristics of nicotine product use have probably changed: children and adolescents are moving away from traditional cigarettes to electronic cigarettes.

In 2021, Worthen and Ahmad reported that social reasons (51.3%) and flavours (41.1%) were the primary reason given for initiating ENDSs [31]. Saddleson et al. conducted a study among New York college students and found that 72.1% of current users reported using ENDSs for enjoyment [32]. In addition, the relatively easy availability of these products on the market probably also plays a significant role [33]. Adolescents often turn to e-cigarettes under the influence of their peers and family members [34]. The results of our study suggest that adolescents choose both traditional cigarettes and e-cigarettes as their first tobacco product almost equally. In the majority of cases, the reason why adolescents reach for products containing nicotine is curiosity, and they also do it under the influence of their peer group. E-cigarettes are gaining popularity among teenagers because liquids are available in various flavours. The commonly chosen flavours are those with a sweet [35], fruity or menthol note [36,37]. The results of our survey are consistent with previous studies, which also indicate similar flavour preferences in this age group.

In addition to a variety of flavours, liquids can also contain vegetable glycerin, propylene glycol, water, and other ingredients that affect their properties, as well as nicotine. In recent scientific reports, most researchers limited their questions in surveys directed to adolescents only to check whether the liquids they use contain nicotine. In the vast majority of cases, adolescents confirm that their liquids contain nicotine [38,39].

In our survey, we went a step further and additionally asked adolescents about the nicotine concentration of the liquids they used. Only 13.8% of the adolescents surveyed said they used nicotine-free liquids. In the majority of cases, like students in the United States [40], their peers in Poland preferred liquids with an average nicotine concentration of 9 to 12 mg/mL in their devices.

The usage patterns and popularity of products may vary across countries, making direct comparisons challenging without considering national regulations. However, it is important to recognize that legal frameworks, market conditions, and public opinion can influence the manner in which these products are used, thereby affecting users’ nicotine and toxic substance exposure [34].

In Poland, there is a minimum age of 18 to purchase tobacco products. Minors are not allowed to purchase or possess cigarettes or other tobacco products. Vendors are required to confirm the age of those who attempt to purchase tobacco products or e-cigarettes, and must refuse to sell if the customer is not of age. It should be noted that it is not possible to purchase e-cigarettes online in Poland.

However, despite these restrictions, adolescents in Poland purchase cigarettes (51.5%) or e-cigarettes (50.6%) on their own in half of the cases, similar to their peers in the United States, who report that they purchased their first electronic nicotine delivery system (ENDS) either independently at a vape shop or online (31.1%), or acquired it from another person (16.3%) [41]. Another study conducted in the United States indicates that youth most frequently obtain e-cigarettes from friends (51.5%), followed by family members (16.4%), vape shops (16.2%), and retail outlets (12.3%) [42]. We should also note the reports suggesting that adolescents in the US are able to purchase e-cigarettes online despite restrictions imposed there [41,43].

Some adolescents also obtain their devices from friends or family members, but in both Poland and the US, the percentage of those who make such purchases through adult intermediaries is much smaller compared to those who purchase the products on their own [44].

Only slightly more than half of current adolescent smokers of traditional cigarettes, as well as a similar number of adolescents who use e-cigarettes, conceal this fact from their parents. It should be noted that when at least one parent is aware that their child is reaching for such products, it is usually the mother. However, there are reasons for concern, as a significant number of adolescents do not conceal their cigarette smoking or e-cigarette use from their parents. This is significant, especially in light of previous studies, which have shown that parents’ lack of or minimal reaction to their child’s smoking or tobacco use may be associated with an increased risk that their child will start using these types of products [45].

The results of our survey show significant associations between demographic factors and e-cigarette use among adolescents.

The males showed an increased risk of using e-cigarettes compared to their female peers, and a similar relationship was also confirmed by other researchers from South Korea [9], Finland [46] and Poland [47]. The higher frequency of e-cigarette use by males may be due to the belief that they are less harmful than traditional cigarettes, while girls are consistently less inclined to perceive e-cigarettes as less harmful than traditional cigarettes compared to their male peers [48].

In our study, we showed that the place of residence (large city > 250,000 residents) influences the increase in the probability of e-cigarette use by children and adolescents. Similar results were obtained in the PolNico study [49] and were reported by researchers from South Korea, who demonstrated a significant association between both current and past e-cigarette use and residency in a large urban area [24]. It seems that one reason for this situation may be the fact that residents of larger cities have better accessibility to e-cigarette points of sale compared to residents of smaller cities or rural towns. Our study also revealed a correlation between attending a secondary technical school and an increase in the risk of e-cigarette use among adolescents. In our survey, we divided secondary schools into comprehensive, technical and vocational ones, with similar divisions of secondary schools rarely found in studies conducted in the United States or Asia. An evaluation of the influence of sociodemographic factors on smoking, e-cigarette use and dual smoking in the population of school children and adolescents in Korea in 2019 also confirmed a higher prevalence of smoking in males and high school students [9]. In a 2022 survey of students from southern Poland, those attending vocational education schools were found to be up to 2.28 times more likely to use e-cigarettes compared to their peers enrolled in general education schools [50]. Another analysis performed on the study group found an association between the education level of mothers and the likelihood of their children and adolescents using e-cigarettes. However, the higher education level of mothers of adolescent e-cigarette users compared to mothers of cigarette-smoking adolescents may be a less clear issue, as in other studies, the authors suggest that higher parental education may act as a protective factor against their children’s e-cigarette use [51]. At the same time, the results of other studies conducted in Poland also indicate that the secondary education of mothers may be a risk factor for adolescent e-cigarette use [49,52]. Our study revealed that higher education of mothers was a risk factor for their children’s e-cigarette use. In contrast, researchers in the Swiss LUIS study conducted in 2022 found no association between the educational levels of mothers and fathers and any smoking behaviour [53]. Divergent findings regarding the relationship between parental educational attainment and the use of e-cigarettes or traditional cigarettes by their children may stem from several factors. Firstly, the cultural and social context plays a pivotal role; research outcomes can vary significantly depending on the cultural background of the country in question. For instance, in Poland, the perception and use of e-cigarettes may differ markedly from that in Switzerland, thereby influencing the research results.

Our research also highlighted the relationship between demographics and the likelihood of using e-cigarettes and traditional cigarettes simultaneously. It turned out that the use of both types of tobacco products simultaneously, so-called “dual smoking”, is more common among younger children than among teenagers, who are more likely to use one type of product.

Another important result of our study is that parents of adolescents who use e-cigarettes exclusively are less likely to use any tobacco products. And if the parents use any tobacco products, their children are significantly less likely to use traditional cigarettes, compared to the group of children of parents who only smoke traditional cigarettes. It suggests that adolescents who use e-cigarettes are less likely to inhale tobacco smoke passively in their home. It must be noted that this relationship is not clear, as some studies confirm the association between exposure to tobacco smoke in the home environment and e-cigarette use [25,54]. On the other hand, there are also reports that do not confirm the above association among the adolescent population, which is consistent with the results of our study [55].

Adolescents who use e-cigarettes exclusively show greater regularity in exercising than their peers who smoke traditional cigarettes or use both types of tobacco products at the same time. This is most likely due to the fact that the choice of e-cigarettes as an alternative among adolescents is often linked to the idea of a healthy lifestyle. Adolescents who choose e-cigarettes often believe that they are less harmful than traditional cigarettes.

## 6. Limitations

Our paper is based on a study conducted through a survey, filled in by a large sample of schoolchildren in Poland. It provides a valuable addition to the scientific literature on adolescent tobacco use, focusing mainly on the analysis of demographic variables and habits related to smoking tobacco cigarettes, using e-cigarettes and using both products simultaneously.

Although the large sample of school-aged adolescents in Poland and the robust study design strengthen our findings, some limitations inherent to the methodology persist.

The data were obtained by using the method of sending an online survey to schools to reach school-aged adolescents and obtain the desired study group. Despite the effectiveness of this method, taking into account the large number of adolescents, the sample was limited to a select subset of schools in Poland. As a result, the data obtained may not fully reflect the characteristics of all adolescents in the country and thus may not be representative of the general population. Access to students was restricted due to the COVID-19 pandemic. The survey was distributed by teachers and not all students had the opportunity to participate. Technical issues, such as lack of internet access when the survey was available, along with students’ absences during online classes or unwillingness to take part in the study resulted in a limited number of respondents. Such circumstances could significantly impact the representativeness of the sample, consequently limiting the generalizability of the findings to the entire student population.

The study only included students from selected educational institutions, which constitutes another crucial limiting factor. The survey did not reach all schools for various reasons, including internal regulations of specific educational institutions that prevented the study from being conducted. Additionally, the lack of consent from school administrations and the regional education authorities for the participation of certain institutions in the research also influenced the survey’s reach. The reluctance of teachers to share the survey with students during class time might have further resulted in a smaller number of participants. Finally, the lack of response to inquiries sent to some schools also limited the availability of respondents.

We used a definition of the current use of tobacco products, defining it as use in the past 30 days. Such a definition is commonly used in cross-sectional studies assessing the use of psychoactive substances by adolescents. It should be noted, however, that such definitions have limited effectiveness in identifying patterns of product use that extend beyond experimental experience and may involve varying degrees of health risk.

All the mentioned factors may significantly influence the diversity of the results and restrict the ability to generalize the conclusions to the entire student population. Therefore, it is essential to treat the obtained results with caution and to take these limitations into consideration in further analyses and interpretations.

In future research, it would be valuable to repeat the survey to analyze trends in the use of traditional cigarettes and e-cigarettes, while also incorporating new forms of nicotine delivery, such as nicotine pouches. It is important to focus on assessing the degree of nicotine addiction among adolescents by adapting addiction scales to account for these emerging delivery methods. Additionally, a comprehensive analysis of existing strategies, along with the development of new methods, is essential to mitigate nicotine use among adolescents. This multifaceted approach will enhance the understanding of nicotine consumption patterns and contribute to more effective prevention and intervention strategies. This is especially important given that the method and frequency of nicotine use significantly impact users’ health risks [5,56,57].

## 7. Conclusions

In conclusion, the current study provides important new insights into demographic factors influencing e-cigarette use among adolescents. We have identified demographic predictors (gender, size of the place of residence, school type, mother’s education) associated with the use of specific devices, which can help inform targeted interventions to reduce e-cigarette use among adolescents. By analyzing the above relationships, we can determine that the representative adolescent is one who uses e-cigarettes, is a boy, lives in a larger city, and attends a secondary technical school, whose mother has higher education. This boy exercises more regularly than his classmates who smoke traditional cigarettes and more often than his classmates who use both types of products. His parents are less likely to smoke traditional cigarettes than the parents of those classmates who are smokers but are more likely to use e-cigarettes than the parents of his non-smoking classmates. Further research is needed to assess how tobacco regulatory policies affect changes in adolescents’ use of certain types of e-cigarette devices.

## Figures and Tables

**Table 1 ijerph-21-01493-t001:** Demographics of the tudy group.

Place of residence		
	n	%
rural area	5492	52.9
small town	2573	24.8
medium-sized town	1141	11
big city	1182	11.4
Mother’s education		
	n	%
primary	1035	10
lower secondary	573	5.5
vocational	2111	20.3
secondary	3405	32.8
higher	3114	30
No data	150	1.4
Father’s education		
	n	%
primary	526	5.1
lower secondary	130	1.3
vocational	3717	35.8
secondary	3292	31.7
higher	2538	24.4
No data	185	1.8
School type		
	n	%
primary school	1988	19.1
secondary comprehensive school	3376	32.5
secondary technical school	4233	40.7
vocational school	791	7.6
Do you exercise regularly?		
	n	%
Yes	6421	61.8
No	3967	38.2
Do you exercise during PE lessons?		
	n	%
Yes	9325	89.8
No	1063	10.2
Do you attend a sports school?		
	n	%
Yes	1583	15.2
No	8805	74.8
Do you attend additional sports training sessions?		
	n	%
Yes	2803	27
No	7585	73

**Table 2 ijerph-21-01493-t002:** Demographics of groups using traditional cigarettes and e-cigarettes.

Traditional Cigarettes (n = 1275)			E-Cigarettes (n = 1546)		
Place of residence					
	n	%		n	%
rural area	651	51.1	rural area	713	46.1
small town	332	26	small town	420	27.2
medium-sized town	141	11.1	medium-sized town	203	13.1
big city	151	11.8	big city	210	13.6
Mother’s education					
	n	%		n	%
primary	140	11	primary	129	8.3
lower secondary	91	7.1	lower secondary	128	8.3
vocational	299	23.5	vocational	318	20.6
secondary	422	33.1	secondary	521	33.7
higher	314	24.6	higher	428	27.7
No data	9	0.7	No data	22	1.4
Father’s education					
	n	%		n	%
primary	91	7.1	primary	85	5.5
lower secondary	28	2.2	lower secondary	24	1.6
vocational	494	38.7	vocational	567	36.7
secondary	384	30.1	secondary	490	31.7
higher	264	20.7	higher	355	23
No data	14	1.1	No data	25	1.6
School type					
	n	%		n	%
primary school	65	5.1	primary school	80	5.2
secondary comprehensive school	423	33.2	secondary comprehensive school	531	34.3
secondary technical school	566	44.4	secondary technical school	754	48.8
vocational school	221	17.3	vocational school	181	11.7
Do you exercise regularly?					
	n	%		n	%
Yes	659	51.7%		874	56.5%
No	616	48.3%		672	46.5%
Do you exercise during PE lessons?					
	n	%		n	%
Yes	1057	82.9%		1309	84.7%
No	218	17.1%		237	15.5%
Do you attend a sports school?					
	n	%		n	%
Yes	194	15.2%		267	17.3%
No	1081	84.8%		1279	82.7%
Do you attend additional sports training sessions?					
	n	%		n	%
Yes	325	15.5%		437	28.3%
No	950	84.5%		1109	71.7%

**Table 3 ijerph-21-01493-t003:** Comparison of how minors and adults purchase traditional cigarettes.

		Minors (<18)	Adults (≥18 and <19)
Someone offers me a cigarette	*N*	72	34
	%	9.80%	6.30%
I buy cigarettes from another person	*N*	8	5
	%	1.10%	0.90%
I ask someone to give me a cigarette	*N*	47	12
	%	6.40%	2.20%
I ask someone to buy me cigarettes	*N*	138	8
	%	18.70%	1.50%
I buy cigarettes from the store myself	*N*	380	466
	%	51.50%	86.80%
Another method	*N*	93	12
	%	12.60%	2.20%

**Table 4 ijerph-21-01493-t004:** Comparison of how minors and adults purchase e-cigarettes.

		Minors (<18)	Adults (≥18 and <19)
I give someone money to buy them for me	*N*	93	14
	%	10.90%	2.00%
Someone offers me an e-cigarette	*N*	117	66
	%	13.80%	9.50%
I buy e-cigarettes from another person	*N*	82	30
	%	9.60%	4.30%
I buy them from the store myself	*N*	430	556
	%	50.60%	79.90%
Another method	*N*	128	30
	%	15.10%	4.30%

**Table 5 ijerph-21-01493-t005:** Demographic variables versus tobacco use.

		Does Not Use Tobacco Products	Traditional Cigarettes	E-Cigarettes	Dual Users	
		(n = 8234)	(n = 608)	(n = 879)	(n = 667)	
Sex	Female	4691	321	418	348	χ^2^(3) = 34.43
		57.00%	52.80%	47.60%	52.20%	*p* < 0.001
	Male	3543	287	461	319	*V* = 0.06
		43.00%	47.20%	52.40%	47.80%	
Place of residence	rural area	4459	320	382	331	*H*(3) = 42.19
		54.20%	52.60%	43.50%	49.60%	*p* < 0.001
	small town	1999	154	242	178	
		24.30%	25.30%	27.50%	26.70%	
	medium-sized town	873	65	127	76	
		10.60%	10.70%	14.40%	11.40%	
	big city	903	69	128	82	
		11.00%	11.30%	14.60%	12.30%	
Mother’s education	primary	840	66	55	74	*H*(3) = 26.7
		10.40%	10.90%	6.40%	11.20%	*p* < 0.001
	lower secondary	407	38	75	53	
		5.00%	6.30%	8.70%	8.00%	
	vocational	1644	149	168	150	
		20.30%	24.70%	19.50%	22.70%	
	secondary	2666	218	317	204	
		32.90%	36.10%	36.80%	30.80%	
	higher	2553	133	247	181	
		31.50%	22.00%	28.70%	27.30%	
Father’s education	primary	394	47	41	44	*H*(3) = 38.91
		4.90%	7.80%	4.80%	6.70%	*p* < 0.001
	lower secondary	92	14	10	14	
		1.10%	2.30%	1.20%	2.10%	
	vocational	2899	251	324	243	
		35.90%	41.80%	37.60%	36.80%	
	secondary	2613	189	295	195	
		32.30%	31.40%	34.30%	29.50%	
	higher	2083	100	191	164	
		25.80%	16.60%	22.20%	24.80%	
School type	primary school	1889	19	34	46	χ^2^(3) = 541.46
		22.90%	3.10%	3.90%	6.90%	*p* < 0.001
	secondary comprehensive school	2649	196	304	227	*V* = 0.13
		32.20%	32.20%	34.60%	34.00%	
	secondary technical school	3203	276	464	290	
		38.90%	45.40%	52.80%	43.50%	
	vocational school	493	117	77	104	
		6.00%	19.20%	8.80%	15.60%	
Participation in PE lessons	no	732	94	113	124	χ^2^(3) = 91.56
		8.90%	15.50%	12.90%	18.60%	*p* < 0.001
	yes	7502	514	766	543	*V* = 0.09
		91.10%	84.50%	87.10%	81.40%	
Regular exercise	no	2995	300	356	316	χ^2^(3) = 69.38
		36.40%	49.30%	40.50%	47.40%	*p* < 0.001
	yes	5239	308	523	351	*V* = 0.08
		63.60%	50.70%	59.50%	52.60%	
Sports school	no	6999	527	725	554	χ^2^(3) = 7.13
		85.00%	86.70%	82.50%	83.10%	*p* = 0.068
	yes	1235	81	154	113	
		15.00%	13.30%	17.50%	16.90%	
Participation in sports training sessions	no	6017	459	618	491	χ^2^(3) = 5.30
		73.10%	75.50%	70.30%	73.60%	*p* = 0.151
	yes	2217	149	261	176	
		26.90%	24.50%	29.70%	26.40%	
Any form of parents’ smoking	no	6917	401	642	457	χ^2^(3) = 241.77
		84.00%	66.00%	73.00%	68.50%	*p* < 0.001
	yes	1317	207	237	210	*V* = 0.15
		16.00%	34.00%	27.00%	31.50%	
Smoking of traditional cigarettes by parents	no	6917	401	642	457	χ^2^(3) = 233.77
		84.00%	66.00%	73.00%	68.50%	*p* < 0.001
	yes	1317	207	237	210	*V* = 0.15
		16.00%	34.00%	27.00%	31.50%	
Use of traditional cigarettes by parents	no	8141	599	854	627	χ^2^(3) = 101.70
		98.90%	98.50%	97.20%	94.00%	*p* < 0.001
	yes	93	9	25	40	*V* = 0.10
		1.10%	1.50%	2.80%	6.00%	

## Data Availability

The original contributions presented in the study are included in the article; further inquiries can be directed to the corresponding author.

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
