# Peer review of "Epidemiology of Traditional Cigarette and E-Cigarette Use Among Adolescents in Poland: Analysis of Sociodemographic Risk Factors"

_ijerph, 2024, doi:10.3390/ijerph21111493_

Round 1

Reviewer 1 Report

Comments and Suggestions for Authors

Thank you for the opportunity to review this manuscript. This is a novel and highly topical research, conducted within a large population of students in Poland. To increase the impact of this publication and to reflect the high volume of work that clearly went into it, I think the Authors should consider the following:

-          There is no analysis section, which I assume is an oversight, but it is necessary to describe how data were analysed.

-          Results section could be much more concise. Please consider reorganising the way in which the results are presented – there are several tables, followed by text describing each table in detail. This makes this section hard to follow and pick the key findings. I suggest combining some tables/moving some tables into appendices, to avoid repeating in text all the findings presented in tables (describe the key findings only). Please signpost tables at the start of the paragraph in which the results in that table are presented.

-          The discussion covers several very interesting findings; I felt they needed to be better grounded and discussed in the context of existing research, with some interpretation of findings. As it is, it reads more like a description of findings, rather than a discussion. (Why could this be, why is this finding important, how does it compare to findings from other years, is this). There is some evidence of this, but each key finding could be discussed in context.

-          Please strengthen the conclusions of the study. It would be good to see some conclusions relating to trends of cigarette smoking vs e-cigarette use and how this research compares to other studies, in the past and/or in other countries.

-         

-          Please proofread the whole text very carefully – I found one or two notes to self, such as “add your calculations here” (line 389) etc. Avoid repetitiveness, this could be more concise, with key points highlighted.  

Specific comments

Line 61 – “e-cigarette use” rather than “smoking” (please check the whole manuscript for this phrase)

Line 89 – please could the authors specify what online method was used to adapt the survey?

Please be consistent in the way percentages are calculated (e.g.:  (line 198-199) 325 respondents = 25.5% but in (Table 2) 325 respondents = 15.5% - this is just an example – please check all calculations and specify if presenting percentage of the whole sample/sample of smoker/sample of current smokers etc.

Comments on the Quality of English Language

 Please avoid the conversational language throughout the paper – please use more uniform terms (e.g.:  the respondents are called adolescents, teens, children, students, respondents, people etc; the authors are called the scientists, surveyors, researchers etc. Using consistent terms would help the reader follow the text better. Please consider making the text easier to read by reducing the use of conversational phrases such as “at the same time” and presenting necessary data as (n=xxx, x%) instead of (xxx people, x%). I would encourage the authors to have the manuscript proofread for academic language.

Author Response

Dear Editor and Reviewers,

We would like to express our gratitude for your thorough review of our manuscript. We have carefully considered all the comments and suggestions the three reviewers provided and made the necessary revisions to enhance the quality of our work.

All changes made in the manuscript are highlighted in red, with original text struck through and new information added in red for clarity. Additionally, our detailed responses to the reviewers' comments are included in the attached document.

Thank you once again for your invaluable feedback and support throughout this process. We believe that the revisions have strengthened our manuscript, and we look forward to your further comments.

Sincerely,

Paulina KurdyÅ›-Bykowska

Reviewer 2 Report

Comments and Suggestions for Authors

The purpose of the paper was aimed to assess the prevalence of traditional cigarette and e-cigarette use among adolescents aged 12-18 in Poland, while identifying demographic risk factors associated with their usage.

The study presents very important results in relation to the consumption of combustible cigarettes and other tobacco products, but the category of dual users that is included in the coding of the results is relevant since this group is not regularly presented.

Furthermore, the analysis of risk factors associated with consumption allows design prevention and treatment strategies for this population. However, it is suggested:

- Organize the information so that what is presented in the textand in the tables is not repetitive.

- In the discussion, include proposals for future research and include other limitations of the study, in addition to the sample size.

Author Response

(The authors gave the same response as above.)

Reviewer 3 Report

Comments and Suggestions for Authors

Lines 37-38: I assume you mean the simultaneous use of ENDS and 'smoked tobacco products', if this is what you mean please amend.

Lines 44-45: I am not sure which research work in being referred to here, is it this study you are conducting, and you are stating this as one of your aims, or are you referring to some other previous study? Please clarify

Lines 75-76: Please note that the latest GYTS conducted in Poland was in 2022, it enrolled >3900 school students 13-15, following very robust methodology and the survey included the use of ENDS among those students. Given that GYTS is a large-scale nation-wide study that implements global standardized methodology, please give a good rationale for carrying out your study and what gaps have you identified in the GYTS 2022 which your study will bridge?

https://cdn.who.int/media/docs/default-source/ncds/ncd-surveillance/data-reporting/poland/poland_-2022_gyts_factsheet_508.pdf?sfvrsn=40b48a2a_1&download=true

Lines 87-89: Have you conducted a pilot testing, validity and reliability to this self-designed questionnaire before implementing it at larger scale? Please elaborate on how this questionnaire was developed and reliability, validity methods pursued.

Lines 223-225: Your study title, aim and study population concerns adolescents, and your questionnaire was distributed through their schoolteachers, it is not clear to me how 'adults' >18 years of age were included in this study?

Line 324: I am a bit sceptical about using the term 'affect', I would rather say: no correlation or no association.

Discussion: How would you explain the notable discrepancy between prevalence rates of tobacco use in all forms which is elaborated in your study as compared to results depicted in the GYTS-Poland 2022?

Conclusion 449: I am not sure you can consider an association between smoking prevalence, habits, etc among adolescents with parental smoking habits or occupation as 'predictors', it is precisely just an association to some variables.

Author Response

(The authors gave the same response as above.)

Round 2

Reviewer 3 Report

Comments and Suggestions for Authors

In order to be able to utilize a translated questionnaire, you need to run certain methods for translation-back translation, validity and reliability of the translated version. You should also obtain consent from the original authors. Though I have requested that you need to provide methods and statistical analysis with respect to the translation of the questionnaire in your section on methodology, yet you have not depicted that. Please clarify this in your methods

Comments on the Quality of English Language

Acceptable.

Author Response

Dear Reviewer,

Thank you for your insightful comments.

Following your revision we decided to change line 96 and substitute it with the following sentence: "After analyzing the PATH (the Population Assessment of Tobacco and Health) study and its findings, we independently developed a self-designed questionnaire that was used in this study. ".

In response to your comment on the methods and statistical analysis of the questionnaire, we have added a paragraph detailing the pilot phase and validation of the questions we developed.

"The survey was initially provided to a selection of schools for the 24-hour pilot, during which we collected 700 responses. This effort allowed the authors to move into the validation stage, following a carefully structured methodology to ensure the survey instrument’s reliability and validity. Our first step was to confirm face validity, ensuring the survey items were relevant and appropriately aligned with the constructs we intended to measure. Conducting a pilot test allowed us to gauge the clarity of the questions, identify any areas of confusion, and address ambiguities within the response options. Once we collected the pilot test results, we cleaned the data to remove incomplete or inconsistent responses, which provided a solid foundation for further analysis. We then ran statistical tests, including Principal Components Analysis (PCA), to identify any underlying dimensions and assess the suitability of each item in capturing our intended constructs. To further evaluate the internal consistency of the survey, we calculated Cronbach’s alpha (α > 0,7). This metric gave us insights into how well the questions within each section worked together to measure a cohesive concept. Based on the findings from PCA and Cronbach’s alpha, we revised certain survey items to strengthen their reliability and validity. By following this thorough validation process, we aimed to enhance the quality of the data collected, increase our confidence in interpreting the findings, and extend them to a broader student population. This systematic approach not only strengthens the credibility of the research but also lays a solid groundwork for the subsequent analyses and conclusions drawn from the survey."

All added content was marked with red color in the manuscript.

We trust that this revision meets the expectations of the esteemed Reviewer.

We sincerely appreciate your comments, which have contributed to enhancing the quality of our work and supporting its potential for publication in your journal.

Regards,

Paulina KurdyÅ›-Bykowska